# How Hessian structure explains mysteries in Sharpness regularization

## ABSTRACT

Recent work has shown that first order methods like SAM which implicitly penalize second order information can improve generalization in deep learning. Seemingly similar methods like weight noise and gradient penalties often fail to provide such benefits. We show that these differences can be explained by the structure of the Hessian of the loss. First, we show that a common decomposition of the Hessian can be quantitatively interpreted as separating the feature exploitation from feature exploration. The feature exploration, which can be described by the Nonlinear Modeling Error matrix (NME), is commonly neglected in the literature since it vanishes at interpolation. Our work shows that the NME is in fact important as it can explain why gradient penalties underperform for certain architectures. Furthermore, we provide evidence that challenges the long held equivalence of weight noise and gradient penalties. This equivalence relies on the assumption that the NME can be ignored, which we find does not hold for modern networks since they involve significant feature learning. Intriguingly, we find that regularizing feature exploitation but not feature exploration yields performance comparable to SAM. This suggests that properly controlling regularization on the two parts of the Hessian is important for the success of many second order methods.

## 1 INTRODUCTION

There is a long history in machine learning of trying to use information about the loss landscape geometry to improve gradient-based learning. This has ranged from attempts to use the Fisher information matrix to improve optimization (Martens & Grosse, 2015), to trying to regularize the Hessian to improve generalization (Moosavi-Dezfooli et al., 2019). More recently, first order methods which implicitly use or penalize second order quantities have been used successfully, including the *sharpness aware minimization* (SAM) algorithm (Foret et al., 2020). On the other hand, there are many approaches to use second order information which once seemed promising but have had limited success (Dean et al., 2012). These include methods like weight noise (An, 1996) and gradient norm penalties, which have shown mixed success.

Part of the difficulty of using second order information is the difficulty of working with the Hessian of the loss. With the large number of parameters in deep learning architectures, as well as the large number of datapoints, many algorithms use stochastic methods to approximate statistics of the Hessian Martens & Grosse (2015); Liu et al. (2023). However, there is a *conceptual* difficulty as well which arises from the complicated structure of the loss Hessian itself. Methods development often involves approximating the Hessian via the Gauss-Newton (GN) matrix - which is PSD for convex losses. The indefinite part of the Hessian is often neglected, in part due to the complexity of both its eigenstructure and computation.

In this work we show that it is important to consider *both* parts of the Hessian in order to design good second order methods. We show that the GN part of the Hessian is related to *exploiting* existing linear structure, while the indefinite part of the Hessian, which we dub the *Nonlinear Modeling Error matrix* (NME), is related to *exploring* the effects of switching to different linear regions. We show that the NME depends heavily on the choice of activation, and in particular the second derivative of that activation function. This suggests that second order methods may be more sensitive to the choice of activation functions than first order methods, and that second order methods might be improved by deliberately choosing which parts of the Hessian to penalize.

We then use our theoretical insights to guide experiments which show the following:

- We explain the inconsistencies between the success of SAM and failure of gradient penalty regularizers in certain architectures to the choice of activation functions, and rescue the performance of the gradient penalty by switching ReLU to GELU. To our knowledge we are the first to show that methods using second order information are more sensitive to the choice of activation function.
- We show that weight noise does not perform as well as the gradient penalty it is thought to approximate. We provide evidence that this is due to the analysis neglecting the important effect of the NME matrix, which weight noise implicitly penalizes.
- Furthermore, we show that penalizing the GN part of the Hessian directly while ignoring the Nonlinear Modeling Error does seem to improve generalization.

We conclude with a discussion about how these insights might be used to design activation functions not with an eye towards forward or backwards passes (Pennington et al., 2017; Martens et al., 2021), but for compatibility with second order methods (implicit or explicit).

## 2 Understanding the structure of the Hessian

The key hypothesis of this paper is that the structure of the Hessian can be used to explain the empirical phenomena of Sections 4 and 5. In this section, we lay the ground work by explaining this structure. Given a model $\mathbf{z}(\boldsymbol{\theta}, \mathbf{x})$ defined on parameters $\boldsymbol{\theta}$ and input $\mathbf{x}$, and a loss function $\mathcal{L}(\mathbf{z}, \mathbf{y})$ on the model outputs and labels $\mathbf{y}$, we can write the gradient of the training loss with respect to $\boldsymbol{\theta}$ as

$$\nabla_{\boldsymbol{\theta}} \mathcal{L} = \mathbf{J}^{\mathrm{T}}(\nabla_{\mathbf{z}} \mathcal{L}) \tag{1}$$

where the Jacobian $\mathbf{J} \equiv \nabla_{\boldsymbol{\theta}} \mathbf{z}$. The Hessian $\nabla_{\boldsymbol{\theta}}^2 \mathcal{L}$ can be decomposed as:

$$\nabla_{\boldsymbol{\theta}}^2 \mathcal{L} = \underbrace{\mathbf{J}^{\mathrm{T}} \mathbf{H}_{\mathbf{z}} \mathbf{J}}_{\text{GN}} + \underbrace{\nabla_{\mathbf{z}} \mathcal{L} \cdot \nabla_{\boldsymbol{\theta}}^2 \mathbf{z}}_{\text{NME}} \tag{2}$$

where $\mathbf{H}_{\mathbf{z}} \equiv \nabla_{\mathbf{z}}^2 \mathcal{L}$. The first term is often called the Gauss-Newton (GN) part of the Hessian (Jacot et al., 2020; Martens, 2020). If the loss function is convex with respect to the model outputs/logits (such as for MSE and CE losses), then the GN matrix is positive semi-definite. This term often contributes large eigenvalues. The second term to our knowledge does not have a name so we call it the *Nonlinear Modeling Error* matrix (NME). It is in general indefinite and vanishes to zero at an interpolating minimum $\boldsymbol{\theta}^*$ where the model "fits"the data ($\nabla_z \mathcal{L}(\boldsymbol{\theta}^*) = \mathbf{0}$), as can happen in overparameterized settings. Due to this, it is quite common for studies to drop this term entirely when dealing with the Hessian. For example, many second order optimizers approximate the Hessian $\nabla_{\boldsymbol{\theta}}^2 \mathcal{L}$ with only the Gauss-Newton term (Martens & Sutskever, 2011; Liu et al., 2023). It is also common to neglect this term in theoretical analysis of the Hessian $\nabla_{\boldsymbol{\theta}}^2 \mathcal{L}$ (Bishop, 1995; Sagun et al., 2017). However, we will show why this term should not be ignored.

While the NME term can become small late in training, it encodes significant information during training. More precisely, *it is the only part of Hessian that contains second order information from the model features* $\nabla_{\boldsymbol{\theta}}^2 \mathbf{z}$. The GN matrix only contains second order information about the loss w.r.t. the logits with the term $\mathbf{H}_{\mathbf{z}}$. All the information about the model function in the GN matrix is first-order. In fact, the GN matrix can be seen as the Hessian of an approximation of the loss where a first-order approximation of the model $\mathbf{z}(\boldsymbol{\theta}', \mathbf{x}) \approx \mathbf{z}(\boldsymbol{\theta}, \mathbf{x}) + \mathbf{J}\boldsymbol{\delta}$ ($\boldsymbol{\delta} = \boldsymbol{\theta}' - \boldsymbol{\theta}$) is used (Martens & Sutskever, 2011)

$$\nabla_{\boldsymbol{\delta}}^2 \mathcal{L}(\mathbf{z}(\theta, \mathbf{x}) + \mathbf{J}\boldsymbol{\delta}, \mathbf{y})|_{\theta'=\theta} = \mathbf{J}^{\mathrm{T}} \mathbf{H}_{\mathbf{z}} \mathbf{J} \tag{3}$$

Thus we can see the GN matrix as the result of a linearization of the model and the NME as the part that takes into account the non-linear part of the model. The GN matrix exactly determines the linearized (NTK) dynamics of training, and therefore controls learning over small parameter changes when the features can be approximated as fixed (see Appendix A.1). In contrast, the NME encodes information about the *changes* in the NTK (Agarwala et al., 2022). For example given a piecewise defined loss surface, we can think of the GN part of the Hessian as *exploiting* the linear (NTK) structure, while the NME gives information on *exploration* - namely, the benefits of switching to a

different region (and thus modifying the NTK). See Figure 1 for an illustration of this with ReLU model. We discuss this aspect further in Section 4.4.

The GN part may *seem* like it must contain this second order information due to its equivalence to the Fisher information matrix for losses that can be written as negative log-likelihoods, like MSE and cross-entropy. For these, the Fisher information itself can be written as the Hessian of a slightly different loss (Pascanu & Bengio, 2013):

$$\mathbf{F} = \mathrm{E}_{\hat{\mathbf{y}} \sim \mathbf{p_z}} \left[ \nabla_{\boldsymbol{\theta}}^2 \mathcal{L}(\mathbf{z}, \hat{\mathbf{y}}) \right] \tag{4}$$

where the only difference is that the labels $\hat{\mathbf{y}}$ are sampled from the model instead of the true labels. However, the NME is 0 for this loss. For example, in the case of MSE using Equation 2 we have

$$\mathrm{E}_{\hat{\mathbf{y}} \sim \mathbf{p_z}} \left[ \nabla_{\boldsymbol{\theta}}^2 \mathcal{L}(\mathbf{z}, \hat{\mathbf{y}}) \right] = \mathrm{E}_{\hat{\mathbf{y}} \sim \mathcal{N}(\mathbf{z}, \mathbf{I})} \left[ \mathbf{J}^\mathrm{T} \mathbf{H_z} \mathbf{J} + \nabla_{\mathbf{z}} \mathcal{L}(\mathbf{z}, \hat{\mathbf{y}}) \cdot \nabla_{\boldsymbol{\theta}}^2 \mathbf{z} \right] \tag{5}$$

$$= \mathbf{J}^\mathrm{T} \mathbf{H_z} \mathbf{J} + \underbrace{\mathrm{E}_{\hat{\mathbf{y}} \sim \mathcal{N}(\mathbf{z}, \mathbf{I})} [\mathbf{z} - \hat{\mathbf{y}}]}_{} \cdot \nabla_{\boldsymbol{\theta}}^2 \mathbf{z} \tag{6}$$

The second term in Equation 6 (NME) vanishes because we are at the global minimum for this loss.

## 2.1 EFFECT OF ACTIVATION FUNCTIONS ON THE NME

One important feature of the NME is that it depends heavily on the choice of activation function - and in particular the second derivatives of the activation function. This means that for activations like ReLU with poorly-defined second derivatives, pointwise computation of the NME can fail to capture the effects of taking activations in and out of saturation.

Given an activation function $\phi$, a feedforward network with $L$ layers on an input $\mathbf{x}_0$ defined iteratively by

$$\mathbf{h}_l = \mathbf{W}_l \mathbf{x}_l, \ \mathbf{x}_{l+1} = \phi(\mathbf{h}_l) \tag{7}$$

The gradient of the model output $\mathbf{x}_L$ with respect to a weight matrix $\mathbf{W}_l$ is given by

$$\frac{\partial \mathbf{x}_L}{\partial \mathbf{W}_l} = \mathbf{J}_{L(l+1)} \circ \phi'(\mathbf{h}_l) \otimes \mathbf{x}_l, \ \mathbf{J}_{l'l} \equiv \prod_{m=l}^{l'-1} \phi'(\mathbf{h}_m) \circ \mathbf{W}_m \tag{8}$$

where $\circ$ is the Hadamard (elementwise) product. The second derivative can be written as:

$$\frac{\partial^2 \mathbf{x}_L}{\partial \mathbf{W}_l \partial \mathbf{W}_m} = \left[ \frac{\partial \mathbf{J}_{L(l+1)}}{\partial \mathbf{W}_m} \circ \phi'(\mathbf{h}_l) + \mathbf{J}_{L(l+1)} \circ \frac{\partial \phi'(\mathbf{h}_l)}{\partial \mathbf{W}_m} \right] \otimes \mathbf{x}_l \tag{9}$$

where without loss of generality $m \geq l$. The full analysis of this derivative can be found in Appendix A.2. The key feature is that the majority of the terms have a factor of the form

$$\frac{\partial \phi'(\mathbf{h}_o)}{\partial \mathbf{W}_m} = \phi''(\mathbf{h}_o) \circ \frac{\partial \mathbf{h}_o}{\partial \mathbf{W}_m} \tag{10}$$

via the product rule - a dependence on $\phi''$. On the diagonal $m = l$, all the terms depend on $\phi''$. We note that a similar analysis can be found in Section 8.1.2 of Martens (2020).

Therefore the second derivative of the activation function is key to controlling the statistics of the NME. Due to the popularity of first order optimizers, activation functions have been designed to have well behaved first derivatives, but not second derivatives. For example, ReLU became popular as a way to deal with gradient propagation issues from activations like $\tanh$; however, it suffers from a "missing curvature" phenomenology - the ReLU second derivative is 0 everywhere except the origin, where it is undefined. This implies that the diagonal of the NME matrix is 0 for ReLU almost everywhere. We will discuss the implications of this dependence more in Section 4.4.

## 3 EXPERIMENTAL SETUP

Our analysis of the Hessian begs an immediate question: when does the NME affect learning algorithms? We conducted experimental studies to answer this question in the context of curvature regularization algorithms which seek to promote convergence to flat areas of the loss landscape. We use the following two setups for the remainder of the paper:

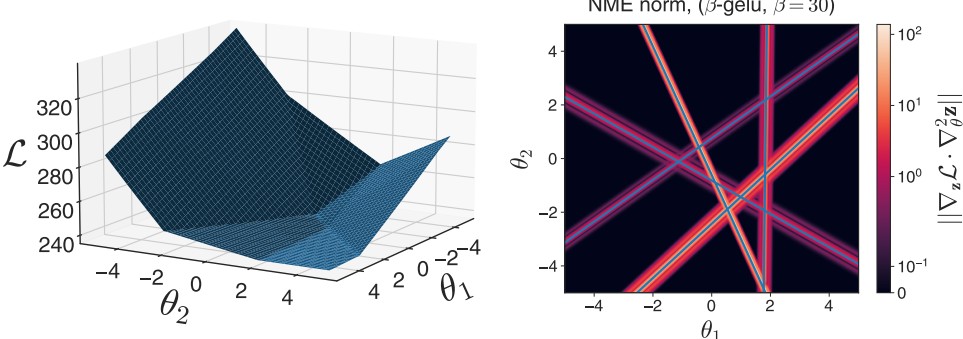

Figure 1: Loss (left) and Nonlinear Modeling Error matrix (NME) norm (right) as a function of 2 parameters in the same hidden layer of an MLP (MSE loss, one datapoint). For ReLU activation model is piecewise multilinear, and piecewise linear for parameters in same layer. Loss is piecewise quadratic for parameters in same layer (left). There is little NME information accessible pointwise and the main features are the boundaries of the piecewise linear regions (blue, right). For $\beta$-GELU, NME magnitude is high only within distance $1/\beta$ of those boundaries. Therefore the NME encodes information about the utility of switching between piecewise multilinear regions.

**Imagenet** We conduct experiments on the popular Imagenet dataset (Deng et al., 2009). All experiments use the Resnet-50 architecture with the same setup and hyper-parameters as Goyal et al. (2018), except that we use cosine learning rate decay (Loshchilov & Hutter, 2016) over 300 epochs.

**CIFAR-10** We also provide results on the CIFAR-10 dataset (Krizhevsky et al., 2009). All experiments use the Resnet-18 architecture with the same setup and hyper-parameters as He et al. (2016), except for the use of cosine learning rate decay.

## 4 PITFALLS FOR SEEKING FLAT MINIMA VIA GRADIENT PENALTY

In this section we leverage our argument about the critical dependence of the NME on activation functions to explain why despite the link between gradient penalty regularizers (Barrett & Dherin, 2021; Smith et al., 2021; Du et al., 2022; Zhao et al., 2022; Reizinger & Huszár, 2023) and Sharpness Aware Minimization (SAM) (Foret et al., 2020), there are conditions in which one performs well but not the other.

### 4.1 SAM

The ideas behind the SAM algorithm originates from seeking a minimum with a *uniformly low loss* in its neighborhood (hence flat). This is formulated in Foret et al. (2020) as a minmax problem,

$$\min_{\boldsymbol{\theta}} \max_{\boldsymbol{\epsilon}} \mathcal{L}(\boldsymbol{\theta} + \boldsymbol{\epsilon}) \quad \text{s.t.} \quad \|\boldsymbol{\epsilon}\| \leq \rho. \tag{11}$$

For computational tractability, Foret et al. (2020) approximates the inner optimization by linearizing $\mathcal{L}$ w.r.t. $\boldsymbol{\epsilon}$ around the origin. Plugging the optimal $\boldsymbol{\epsilon}$ into the objective function yields

$$\min_{\boldsymbol{\theta}} \mathcal{L}\Big(\boldsymbol{\theta} + \rho \frac{\nabla_{\boldsymbol{\theta}} \mathcal{L}(\boldsymbol{\theta})}{\|\nabla_{\boldsymbol{\theta}} \mathcal{L}(\boldsymbol{\theta})\|}\Big). \tag{12}$$

To minimize the above by gradient descent, we would need to compute[1]:

$$\frac{\partial}{\partial \boldsymbol{\theta}} \mathcal{L}\Big(\boldsymbol{\theta} + \rho \frac{\mathbf{g}(\boldsymbol{\theta})}{\|\mathbf{g}(\boldsymbol{\theta})\|}\Big) = \Big(\mathbf{I} + \underbrace{\rho \frac{\mathbf{H}}{\|\mathbf{g}\|}\Big(\mathbf{I} - \frac{\mathbf{g}}{\|\mathbf{g}\|} \frac{\mathbf{g}^{\mathrm{T}}}{\|\mathbf{g}\|}\Big)}_{\text{Hessian related term}}\Big) \nabla_{\boldsymbol{\theta}} \mathcal{L}\Big(\boldsymbol{\theta} + \rho \frac{\mathbf{g}}{\|\mathbf{g}\|}\Big), \ \mathbf{g} \equiv \nabla_{\boldsymbol{\theta}} \mathcal{L}(\boldsymbol{\theta}), \ \mathbf{H} \equiv \nabla_{\boldsymbol{\theta}}^2 \mathcal{L}(\boldsymbol{\theta})$$

$$\tag{13}$$

---

[1]In our notation the gradient and Hessian operators $\nabla$ and $\nabla^2$ precede function evaluation, e.g. $\nabla_{\boldsymbol{\theta}} \mathcal{L}(f(\boldsymbol{\theta}))$ means $\big(\frac{\partial}{\partial \boldsymbol{\tau}} \mathcal{L}(\boldsymbol{\tau})\big)_{\boldsymbol{\tau} = f(\boldsymbol{\theta})}$.

This can still be computationally demanding as it involves the computation of a Hessian-vector product $\mathbf{Hg}$. The SAM algorithm drops the Hessian related term in (13) giving the update rule:

$$\boldsymbol{\theta} \leftarrow \boldsymbol{\theta} - \eta \, \nabla_{\boldsymbol{\theta}} \mathcal{L} \left( \boldsymbol{\theta} + \rho \tilde{\mathbf{g}} \right), \; \tilde{\mathbf{g}} \equiv \nabla_{\boldsymbol{\theta}} \mathcal{L}(\boldsymbol{\theta}) / \|\nabla_{\boldsymbol{\theta}} \mathcal{L}(\boldsymbol{\theta})\| \tag{14}$$

for some step-size parameter $\eta > 0$. A related learning algorithm is unnormalized SAM (USAM) with update rule

$$\boldsymbol{\theta} \leftarrow \boldsymbol{\theta} - \eta \, \nabla_{\boldsymbol{\theta}} \mathcal{L} \left( \boldsymbol{\theta} + \rho \mathbf{g} \right), \; \mathbf{g} \equiv \nabla_{\boldsymbol{\theta}} \mathcal{L}(\boldsymbol{\theta}) \tag{15}$$

USAM has similar performance to SAM and is easier to analyze (Agarwala & Dauphin, 2023).

## 4.2 PENALTY SAM

If $\rho$ is very small, we may approximate $\mathcal{L}$ in (12) by its first order Taylor expansion around the point $\rho = 0$ as below.

$$\mathcal{L}_{\text{PSAM}}(\boldsymbol{\theta}) \triangleq \mathcal{L}(\boldsymbol{\theta})_{\rho=0} + \rho \Big( \frac{\partial}{\partial \rho} \mathcal{L} \Big( \boldsymbol{\theta} + \rho \frac{\nabla_{\boldsymbol{\theta}} \mathcal{L}(\boldsymbol{\theta})}{\|\nabla_{\boldsymbol{\theta}} \mathcal{L}(\boldsymbol{\theta})\|} \Big) \Big)_{\rho=0} = \mathcal{L}(\boldsymbol{\theta}) + \rho \Big\langle \nabla_{\boldsymbol{\theta}} \mathcal{L}(\boldsymbol{\theta}), \frac{\nabla_{\boldsymbol{\theta}} \mathcal{L}(\boldsymbol{\theta})}{\|\nabla_{\boldsymbol{\theta}} \mathcal{L}(\boldsymbol{\theta})\|} \Big\rangle \tag{16}$$

$$= \mathcal{L}(\boldsymbol{\theta}) + \rho \, \|\nabla_{\boldsymbol{\theta}} \mathcal{L}(\boldsymbol{\theta})\| \,. \tag{17}$$

Under this approximation, minimizing $\mathcal{L}_{\text{PSAM}}$ amounts to minimizing the loss $\mathcal{L}$ while penalizing its gradient norm. If $\rho$ is not close to zero, then loss landscape of $\mathcal{L}_{\text{PSAM}}$ provides a very poor approximation to that of 12. We refer to this specific gradient penalty as *Penalty SAM* and denote its associated objective function (17) by PSAM. The unnormalized equivalent PUSAM is

$$\mathcal{L}_{\text{PUSAM}}(\boldsymbol{\theta}) \triangleq \mathcal{L}(\boldsymbol{\theta}) + \rho \, \|\nabla_{\boldsymbol{\theta}} \mathcal{L}(\boldsymbol{\theta})\|^2 \,. \tag{18}$$

## 4.3 PENALTY SAM VS ORIGINAL SAM

Figure 2 shows our experimental results comparing PSAM and SAM across two datasets and networks with different activation functions. Surprisingly, we see that PSAM behaves differently between the two activation functions while SAM is insensitive to them. PSAM does not tolerate larger values for $\rho$ with ReLU networks so well as it does with GELU networks. However, SAM performs well with both activation functions and in fact benefits from larger $\rho$. Thus the networks with GELU are able to reach higher accuracies because they can benefit from larger $\rho$.

This difference between the two methods with different activations is unexpected because switching between these activations does not significantly influence the accuracy of networks trained with SGD: both networks reach roughly 76.7 in our experiments on ImageNet with SGD.

The crucial difference between PSAM and SAM (or even with SGD) is that the gradient of PSAM involves *explicit computation of the Hessian-gradient product*. However, SAM obtains the same second order information implicitly by taking steps in the gradient direction away from the current point as per Equation 14. As we will see in the next section, the choice of activation function has a strong effect on pointwise estimates of the Hessian, and we hypothesize that the failure of ReLU networks to surface this higher order information means that PSAM with ReLU has poor information on further away points.

## 4.4 NONLINEAR MODELING ERROR AND ACTIVATION FUNCTIONS

We can study this difference using the $\beta$-GELU which can interpolate between the two activation functions. It is given by

$$\beta\text{-GELU}(x) = x\Phi(\beta x) \tag{19}$$

where $\Phi$ is the standard Gaussian CDF. We can recover GELU by setting $\beta = 1$, and ReLU is recovered in the limit $\beta \to \infty$. The second derivative is given by

$$\frac{d^2}{dx^2} \beta\text{-GELU}(x) = \frac{1}{\sqrt{2\pi\beta^{-2}}} e^{-x^2/2\beta^{-2}} \left[ 2 - (x/\beta^{-1})^2 \right] \tag{20}$$

For large $\beta$, this function is exponentially small when $x \gg \beta^{-1}$, and $O(\beta)$ when $|x| = O(\beta^{-1})$. As $\beta$ increases the non-zero region becomes smaller while the non-zero value becomes larger such

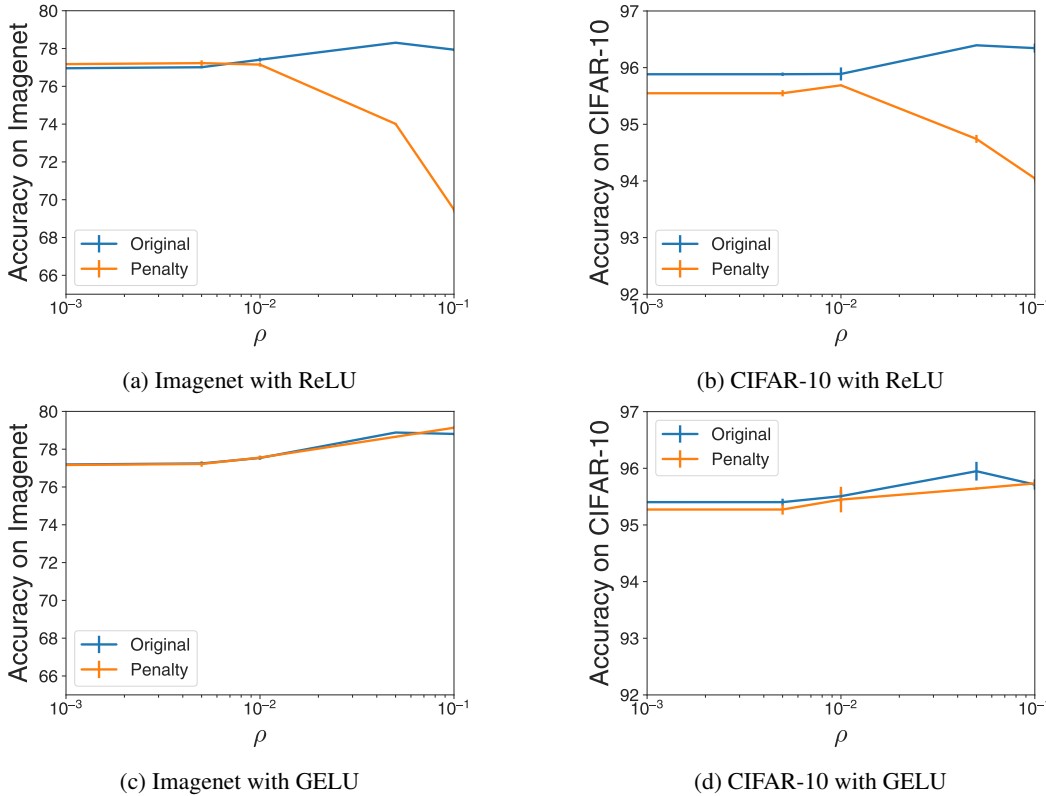

Figure 2: Test Accuracy as $\rho$ increases across different datasets and activation functions averaged over 2 seeds. For ReLU networks and large $\rho$, there is a significant difference between `PSAM` and `SAM`. `PSAM` with GELU networks more closely follows the behavior of `SAM`.

that the integral is always 1. This suggests that rather than being uniformly 0, the ReLU second derivative is better described by Dirac delta "function" (really a distribution) - 0 except at the origin, where it is undefined, but still integrable to 1.

The choice of $\beta$ determines how much information the NME can convey in a practical setting. This second derivative is large only when the input to the activation is within distance $1/\beta$ of 0. In a deep network this corresponds to being near the boundary of the piecewise multilinear regions where the activations switch on and off. We can illustrate this using two parameters of an MLP in the same layer, where the model is in fact piecewise linear with respect to those parameters (Figure 1). The second derivative serves as an "edge detector" (more generally, hyperplane detector), and the NME can be used to probe the usefulness of crossing these edges.

From Equation 9, this means that for intermediate $\beta$ many terms of the diagonal of the NME will be non-zero at a typical point. However as $\beta$ increases, the probability of terms being non-zero becomes low, but when they are non-zero they are large - giving a sparse, spiky structure to the NME, especially on the diagonal. This leads to the NME becoming a high-variance estimator of local structure. Therefore any methods seeking to use this information explicitly are doomed to fail.

Our experiments are consistent with this intuition. In Figure 3, we show that accuracy suffers for penalty `SAM` with larger $\rho$ as we increase $\beta$ but is unaffected for SGD. And in Figure 4 we confirm that $\beta$ effectively controls the sparsity of the activation function Hessian both at initialization and after training. This is evidence that the difference between the different activation functions for penalty `SAM` is explained by the statistics of the NME matrix.

Note that we are not claiming that the choice of the activation function is a sufficient condition for penalty `SAM` to work with larger $\rho$. There are many architectural changes that can affect the NME matrix and we have shown that the statistics of the activation function is a significant one.

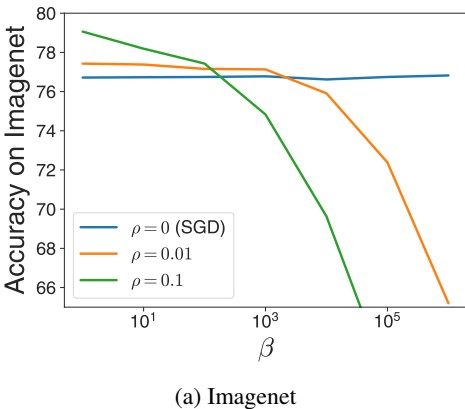
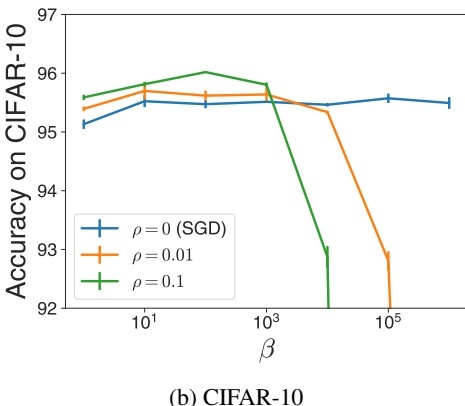

(a) Imagenet
(b) CIFAR-10

Figure 3: Accuracy vs $\beta$ with 3 different settings of $\rho$ for networks with $\beta$-GELU activations (average of 2 seeds). We can see that as the $\beta$-GELU starts to approximates the ReLU accuracy decreases for $\rho > 0$. This effect is more pronounced with larger $\rho$ and is not observe for SGD where $\rho = 0$
.

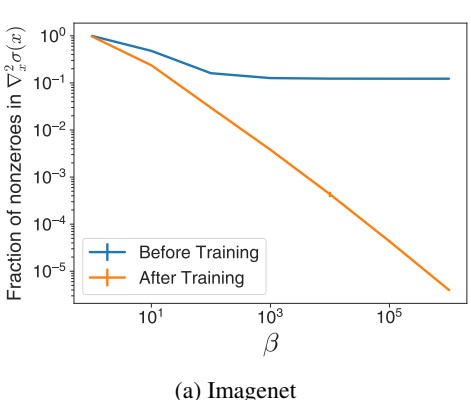
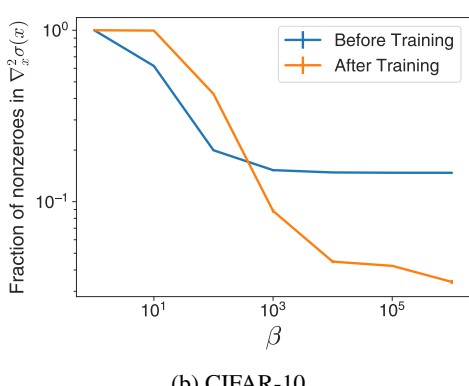

(a) Imagenet
(b) CIFAR-10

Figure 4: Fraction of nonzeroes in $\nabla_x^2 \sigma(x)$ as $\beta$ increases across datasets for networks with $\beta$-GELU activations (average of 2 seeds). We can see that the sparsity of this second derivative increase dramatically as the $\beta$-GELU starts to approximates the ReLU.

## 5 WHY WEIGHT NOISE DOESN NOT WORK

In this section, we investigate why weight noise does not work even though it has long been thought to be equivalent to a gradient penalty similar to 17 (Bishop, 1995). We will see that this connection does not hold for non-linear models due in part to the neglect of the NME term of the Hessian.

### 5.1 WEIGHT NOISE IS SEEN AS A GRADIENT PENALTY

We first review the connection between training with noise and gradient penalty established by Bishop (1995). Though the paper considers input noise, the same analysis can be applied to weight noise. Adding Gaussian $\epsilon \sim \mathcal{N}(0, \sigma^2)$ noise with strength hyper-parameter $\sigma$ to the parameters can be approximated to second order by

$$\mathrm{E}_\epsilon[\mathcal{L}(\boldsymbol{\theta} + \boldsymbol{\epsilon})] \approx \mathcal{L}(\boldsymbol{\theta}) + \underbrace{\mathrm{E}_\epsilon[\nabla_{\boldsymbol{\theta}}\mathcal{L} \cdot \boldsymbol{\epsilon}]} + \mathrm{E}_\epsilon[\boldsymbol{\epsilon}^{\mathrm{T}}\mathbf{H}\boldsymbol{\epsilon}] = \mathcal{L}(\boldsymbol{\theta}) + \sigma^2 \mathrm{tr}(\mathbf{H}) \tag{21}$$

where the second term has zero expectation since $\boldsymbol{\epsilon}$ is mean 0, and the third term is a variation of the Hutchison trace estimator (Hutchinson, 1989). (We note that though the second term vanishes in expectation, it still can have large effects on the training dynamics.) In order to find the connection to gradient penalties, (Bishop, 1995) argues that we can simplify the term related to the Hessian by dropping the NME in Equation 2 for the purposes of minimization

$$\mathrm{tr}(\mathbf{H}) = \mathrm{tr}\left(\mathbf{J}^{\mathrm{T}}\mathbf{H}_{\mathbf{z}}\mathbf{J} + \nabla_{\mathbf{z}}\mathcal{L} \cdot \nabla_{\boldsymbol{\theta}}^2 \mathbf{z}\right) \approx \mathrm{tr}(\mathbf{J}^{\mathrm{T}}\mathbf{H}_{\mathbf{z}}\mathbf{J}) \tag{22}$$

The argument is that for the purposes of training neural networks this term can be dropped because it is zero at the global minimum. For models with mean squared error loss (MSE), this gives us

$$E[\mathcal{L}_{\text{MSE}}(\boldsymbol{\theta} + \boldsymbol{\epsilon})] \approx \mathcal{L}_{\text{MSE}}(\boldsymbol{\theta}) + \sigma^2 \text{tr}(\mathbf{J}^{\text{T}} \mathbf{H}_{\mathbf{z}}^{\text{MSE}} \mathbf{J}) = \mathcal{L}_{\text{MSE}}(\boldsymbol{\theta}) + \sigma^2 \|\nabla_{\boldsymbol{\theta}} \mathbf{z}\|^2 \tag{23}$$

This is strikingly similar to the penalty form of SAM (Equation 17), with the key difference being that it is a gradient penalty on the logits and not the loss. In fact for MSE there is *no* information about the loss in this term.

More recent work has proposed a new estimator for the trace of the Gauss-Newton matrix for cross-entropy loss Wei et al. (2020). Using this estimator, we can express weight noise with cross-entropy loss as

$$\mathcal{L}_{\text{CE}}(\boldsymbol{\theta} + \boldsymbol{\epsilon}) \approx \mathcal{L}_{\text{CE}}(\boldsymbol{\theta}) + \sigma^2 \text{tr}(\mathbf{J}^{\text{T}} \mathbf{H}_{\mathbf{z}}^{\text{CE}} \mathbf{J}) = \mathcal{L}_{\text{CE}}(\boldsymbol{\theta}) + \sigma^2 \mathrm{E}_{\hat{\mathbf{y}} \sim \text{Cat}(\mathbf{z})} \left[ \|\nabla_{\boldsymbol{\theta}} \mathcal{L}(\boldsymbol{\theta}, \hat{\mathbf{y}})\|^2 \right]. \tag{24}$$

This is almost exactly the same as Equation 18, except for the fact that the labels are sampled from the model instead of the ground-truth labels and the norm is squared.

This leads us to two questions: first, what's the difference between a regularizer like Equation 18 and 24? And secondly, is the NME actually negligible? In Section 5.2 we design a series of experiments to probe these questions, and provide evidence that the NME cannot be neglected for modern networks, and there is a difference between penalizing the gradients of $\mathcal{L}$ and penalizing the gradients of $\mathbf{z}$.

## 5.2   WEIGHT NOISE IS NOT EQUIVALENT TO GRADIENT PENALTY

In order to better understand this difference in the two methods, we experimentally evaluated several different variants of the methods to rule out other reasons for the difference in performance between weight noise and SAM/PSAM. All the variants will have the form of an additive penalty

$$\mathcal{L}_{\text{Variant}}(\boldsymbol{\theta}) = \mathcal{L}(\boldsymbol{\theta}) + \rho \Omega_{\text{Variant}}(\boldsymbol{\theta}) \tag{25}$$

The variants we consider are

- Gauss-Newton penalty ($\Omega(\boldsymbol{\theta}) = \mathrm{E}_{\hat{\mathbf{y}} \sim \text{Cat}(\mathbf{z})}[\|\nabla_{\boldsymbol{\theta}} \mathcal{L}(\boldsymbol{\theta}, \hat{\mathbf{y}})\|^2]$): This is the term that using the analysis of (Bishop, 1995) would be equivalent to weight noise (Equation 24). Here we directly penalize this term instead of relying on its regularization through the weight noise, which also penalizes the Nonlinear Modeling Error matrix. We do not pass gradients through the sampling of the labels $\hat{\mathbf{y}}$, but we find similar results if we pass gradients using the straight-through estimator (Bengio et al., 2013). This variant allows to test if training with weight noise is indeed a good approximation of this more costly term.

- Hessian-trace penalty ($\Omega(\boldsymbol{\theta}) = \mathrm{E}_{\boldsymbol{\epsilon} \sim \mathcal{N}(0,1)}[\boldsymbol{\epsilon}^T \mathbf{H} \boldsymbol{\epsilon}]$): This is the term that appears in Equation 21 before we drop the NME term and is an efficient estimator of the trace of the Hessian. This allows to us to single out the second order effect of weight noise, as it's possible the higher order terms from weight noise are detrimental to generalization.

- Unnormalized penalty SAM ($\Omega(\boldsymbol{\theta}) = \|\nabla_{\boldsymbol{\theta}} \mathcal{L}(\boldsymbol{\theta}, \mathbf{y})\|^2$): This is the penalty form of USAM, and allows us to test if the difference between PSAM and weight noise is due to the norm being squared for weight noise.

We only draw a single sample to estimate the expectations in the Gauss-Newton and Hessian-trace penalty. We find the additional hyper-parameter $\rho$ for each method with cross-validation on Imagenet. A smaller hyper-parameter search on CIFAR-10 showed the optimal values to remain stable across the two datasets. For the Gauss-Newton penalty, the grid was $\{10^{-3}, 5 \cdot 10^{-3}, 10^{-2}, 10^{-1}\}$, for Unnormalized PSAM $\{10^{-4}, 10^{-3}, 10^{-2}, 10^{-1}\}$ and $\{10^{-7}, 10^{-6}, 10^{-5}, 10^{-4}\}$ for Hessian-trace (higher values were highly unstable). For weight noise, the standard deviation of the noise is chosen from $\{10^{-3}, 10^{-2}, 10^{-1}\}$.

Table 1 shows that weight noise does not work as well as PSAM on either dataset. On Imagenet, the improvement from PSAM is $+2.2\%$ but it is $+0.3\%$ for weight noise. However, we can see that the performance of the Gauss-Newton is consistently greater than weight noise. It's improvement on Imagenet is a more significant $1.6\%$. The difference between Gauss-Newton and weight noise

| Approach | Penalty form | Imagenet | CIFAR-10 |
|---|---|---|---|
| SGD | | $76.8 \pm 0.0$ | $95.2 \pm 0.1$ |
| Original SAM (Equation 14) | | $78.9 \pm 0.1$ | $95.9 \pm 0.0$ |
| Penalty SAM | $\|\nabla_{\boldsymbol{\theta}}\mathcal{L}(\boldsymbol{\theta},\mathbf{y})\|$ | $79.0 \pm 0.1$ | $95.7 \pm 0.1$ |
| Unnormalized Penalty SAM | $\|\nabla_{\boldsymbol{\theta}}\mathcal{L}(\boldsymbol{\theta},\mathbf{y})\|^2$ | $79.0 \pm 0.0$ | $95.6 \pm 0.0$ |
| Gauss-Newton Penalty | $\mathrm{E}_{\hat{\mathbf{y}}\sim\mathrm{Cat}(\mathbf{z})}\left[\left\|\nabla_{\boldsymbol{\theta}}\mathcal{L}(\boldsymbol{\theta},\hat{\mathbf{y}})\right\|^2\right]$ | $78.4 \pm 0.1$ | $96.2 \pm 0.1$ |
| Hessian-trace Penalty | $\mathrm{E}_{\boldsymbol{\epsilon}\sim\mathcal{N}(0,1)}\left[\boldsymbol{\epsilon}^{\mathrm{T}}\mathbf{H}\boldsymbol{\epsilon}\right]$ | $75.5 \pm 0.3$ | $95.2 \pm 0.1$ |
| Weight Noise | | $77.1 \pm 0.1$ | $95.3 \pm 0.0$ |

Table 1: Accuracy of different penalties averaged over 2 seeds with GELU networks. We can see that weight noise does not match the results of the Gauss-Newton penalty. Instead it roughly matches the results of the Hessian trace penalty, which shows that the Nonlinear Modeling Error term should not be ignored in regularization.

cannot be explained by the first or higher order terms in weight noise since the Hessian-trace penalty does not perform well either. Further, we can see with unnormalized PSAM that squaring the norm has little effect on final accuracy.

These results are evidence that the NME term of the Hessian should not be dropped when applying the analysis of (Bishop, 1995) to weight noise for modern networks. Indeed there is a significant difference between the Hessian trace, which is sensitive to the NME, and the Gauss-Newton penalty, which is not.

## 6 DISCUSSION

Our theoretical analysis gives some understanding of the structure of the Hessian - in particular, the Nonlinear Modeling Error matrix. This piece of the Hessian is often neglected as it is indefinite and tends not to generate large eigenvalues which are the focus of many regularization and optimization efforts. However, the NME can encode important information about nearby regions. It also suggests that pointwise estimates of this function require well-designed activation functions, or else the NME will tend to be sparse.

It is illustrative that SAM is relatively insensitive to the difference between GELU and ReLU, particularly for large $\beta$, but penalty SAM is quite sensitive to the differences. Regular SAM can use the $\rho$-step to effectively "integrate" over the relevant second order information without ever having to compute it explicitly, which makes it less sensitive to architectural choices. This suggests that designing explicit second order methods should involve careful selection, and even tuning, of the activation function second derivatives.

All of our experiments show that the NME should not be neglected when designing second order methods. This applies to methods where second order terms are computed explicitly in the regularizer, those where the regularizer implicitly penalizes second order information as well as first order methods where second order information is computed explicitly while taking gradients. In our experimental setting, we generally found that direct penalization of the NME was detrimental. Similarly, penalty SAM worked worse when the NME information was missing due to the activation function. We hypothesize that this is because the Nonlinear Modeling Error matrix encodes information about loss-relevant feature learning.

## 7 CONCLUSION

Our work sheds light on the complexities of using second order information in deep learning. It is important to consider the effects of *both* the Gauss-Newton and Nonlinear Modeling Error terms, and design algorithms and architectures with that in mind. Designing activation functions for compatibility with second order methods may also be an interesting avenue of future research.

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

# A HESSIAN STRUCTURE

## A.1 GAUSS-NEWTON AND NTK LEARNING

In the large width limit (width/channels/patches increasing while dataset is fixed), the learning dynamics of neural networks are well described by the *neural tangent kernel*, or NTK (Jacot et al., 2018; Lee et al., 2019). Consider a dataset size $D$, with outputs $\mathbf{z}(\boldsymbol{\theta}, \mathbf{X})$ over the inputs $\mathbf{X}$ with parameters $\boldsymbol{\theta}$. The (empirical) NTK $\hat{\boldsymbol{\Theta}}$ is the $D \times D$ matrix given by

$$\hat{\boldsymbol{\Theta}} \equiv \frac{1}{D}\mathbf{J}\mathbf{J}^{\mathrm{T}}, \ \mathbf{J} \equiv \frac{\partial \mathbf{z}}{\partial \boldsymbol{\theta}} \tag{26}$$

For wide enough networks, the learning dynamics can be written in terms of the model output $\mathbf{z}$ and the NTK $\hat{\boldsymbol{\Theta}}$ alone. For small learning rates we can study the gradient flow dynamics. The gradient flow dynamics on the parameters $\boldsymbol{\theta}$ with loss function $\mathcal{L}$ (averaged over the dataset) is given by

$$\dot{\boldsymbol{\theta}} = -\frac{1}{D}\nabla_{\boldsymbol{\theta}}\mathcal{L} = -\frac{1}{D}\mathbf{J}^{\mathrm{T}}\nabla_{\mathbf{z}}\mathcal{L} \tag{27}$$

We can use the chain rule to write down the dynamics of $\mathbf{z}$:

$$\dot{\mathbf{z}} = \frac{\partial \mathbf{z}}{\partial \boldsymbol{\theta}}\dot{\boldsymbol{\theta}} = -\frac{1}{D}\mathbf{J}\mathbf{J}^{\mathrm{T}}\nabla_z\mathcal{L} = -\hat{\boldsymbol{\Theta}}\nabla_z\mathcal{L} \tag{28}$$

In the limit of infinite width, the overall changes in individual parameters become small, and the $\hat{\boldsymbol{\Theta}}$ is fixed during training. This corresponds to the *linearized* or *lazy* regime Chizat et al. (2019); Agarwala et al. (2020). The NTK encodes the linear response of $\mathbf{z}$ to small changes in $\boldsymbol{\theta}$, and the dynamics is closed in terms of $\mathbf{z}$. For finite width networks, this can well-approximate the dynamics for a number of steps related to the network width amongst other properties Lee et al. (2019).

In order to understand the dynamics of Equation 28 at small times, or around minima, we can linearize with respect to $\mathbf{z}$. We have:

$$\frac{\partial \dot{\mathbf{z}}}{\partial \mathbf{z}} = -\frac{\partial \hat{\boldsymbol{\Theta}}}{\partial \mathbf{z}}\nabla_{\mathbf{z}}\mathcal{L} - \hat{\boldsymbol{\Theta}}\mathbf{H}_{\mathbf{z}} \tag{29}$$

where $\mathbf{H}_{\mathbf{z}} = \frac{\partial^2 \mathcal{L}}{\partial \mathbf{z} \partial \mathbf{z}'}$. In the limit of large width, the NTK is constant and the first term vanishes. The local dynamics depends on the spectrum of $\hat{\boldsymbol{\Theta}}\mathbf{H}_{\mathbf{z}}$. From the cyclic property of the trace, the non-zero part of the spectrum is equal to the non-zero spectrum of $\frac{1}{D}\mathbf{J}^{\mathrm{T}}\mathbf{H}_{\mathbf{z}}\mathbf{J}$ - which is the Gauss-Newton matrix.

Therefore the eigenvalues of the Gauss-Newton matrix control the short term, linearized dynamics of $\mathbf{z}$, for fixed NTK. It is in this sense that the Gauss-Newton encodes information about exploiting the local linear structure of the model.

## A.2 NONLINEAR MODELING ERROR AND SECOND DERIVATIVES OF FCNS

We can explicitly compute the Jacobian and second derivative of the model for a fully connected network. We write a feedforward network as follows:

$$\mathbf{h}_l = \mathbf{W}_l\mathbf{x}_l, \ \mathbf{x}_{l+1} = \phi(\mathbf{h}_l) \tag{30}$$

The gradient of $\mathbf{x}_L$ with respect to $\mathbf{W}_l$ can be written as:

$$\frac{\partial \mathbf{x}_L}{\partial \mathbf{W}_l} = \frac{\partial \mathbf{x}_L}{\partial \mathbf{h}_l}\frac{\partial \mathbf{h}_l}{\partial \mathbf{W}_l} \tag{31}$$

which can be written in coordinate-free notation as

$$\frac{\partial \mathbf{x}_L}{\partial \mathbf{W}_l} = \frac{\partial \mathbf{x}_L}{\partial \mathbf{h}_l} \otimes \mathbf{x}_l \tag{32}$$

If we define the partial Jacobian $\mathbf{J}_{l'l} \equiv \frac{\partial \mathbf{x}_{l'}}{\partial \mathbf{x}_l}, l' > l$

$$\frac{\partial \mathbf{x}_L}{\partial \mathbf{W}_l} = \mathbf{J}_{L(l+1)} \circ \phi'(\mathbf{h}_l) \otimes \mathbf{x}_l \tag{33}$$

Here $\circ$ denotes the Hadamart product, in this case equivalent to matrix multiplication by $\mathrm{diag}(\phi'(\mathbf{h}_m))$.

The Jacobian can be explicitly written as

$$\mathbf{J}_{l'l} = \prod_{m=l}^{l'-1} \phi'(\mathbf{h}_m) \circ \mathbf{W}_m \tag{34}$$

Therefore, we can write:

$$\frac{\partial \mathbf{x}_L}{\partial \mathbf{W}_l} = \left[ \prod_{m=l+1}^{L-1} \phi'(\mathbf{h}_m) \circ \mathbf{W}_m \right] \circ \phi'(\mathbf{h}_l) \otimes \mathbf{x}_l \tag{35}$$

The second derivative is more complicated. Consider

$$\frac{\partial^2 \mathbf{x}_L}{\partial \mathbf{W}_l \partial \mathbf{W}_m} = \frac{\partial}{\partial \mathbf{W}_m} \left[ \mathbf{J}_{L(l+1)} \circ \phi'(\mathbf{h}_l) \otimes \mathbf{x}_l \right] \tag{36}$$

for weight matrices $\mathbf{W}_l$ and $\mathbf{W}_m$. Without loss of generality, assume $m \geq l$.

We first consider the case where $m > l$. In this case, we have

$$\frac{\partial \phi'(\mathbf{h}_l)}{\partial \mathbf{W}_m} = 0, \ \frac{\partial \mathbf{x}_l}{\partial \mathbf{W}_m} = 0 \tag{37}$$

since $\mathbf{W}_m$ comes after $\mathbf{h}_l$. If we write down the derivative of $\mathbf{J}_{L(l+1)}$, there are two types of terms. The first comes from the direct differentiation of $\mathbf{W}_m$; the others come from differentation of $\phi'(\mathbf{h}_n)$ for $n \geq m$. We have:

$$\frac{\partial \mathbf{J}_{L(l+1)}}{\partial \mathbf{W}_m} = \mathbf{J}_{L(m+1)} \phi'(\mathbf{h}_m) \frac{\partial \mathbf{W}_m}{\partial \mathbf{W}_m} \mathbf{J}_{(m-1)(l+1)} + \sum_{o=m}^{L-1} \mathbf{J}_{L(o+1)} \frac{\partial \phi'(\mathbf{h}_o)}{\partial \mathbf{W}_m} \mathbf{W}_o \mathbf{J}_{(o-1)(l+1)} \tag{38}$$

The $\mathbf{W}_m$ derivative projected into a direction $\mathbf{B}$ can be written as:

$$\begin{aligned}
\frac{\partial \mathbf{J}_{L(l+1)}}{\partial \mathbf{W}_m} \cdot \mathbf{B} &= \mathbf{J}_{L(m+1)} \phi'(\mathbf{h}_m) \mathbf{B} \mathbf{J}_{(m-1)(l+1)} \\
&+ \sum_{o=m}^{L-1} \mathbf{J}_{L(o+1)} \left( \phi''(\mathbf{h}_o) \circ \mathbf{W}_o \frac{\partial \mathbf{x}_{o-1}}{\partial \mathbf{W}_m} \cdot \mathbf{B} \right) \mathbf{W}_o \mathbf{J}_{(o-1)(l+1)}
\end{aligned} \tag{39}$$

From our previous analysis, we have:

$$\begin{aligned}
\frac{\partial \mathbf{J}_{L(l+1)}}{\partial \mathbf{W}_m} \cdot \mathbf{B} &= \mathbf{J}_{L(m+1)} \phi'(\mathbf{h}_m) \mathbf{B} \mathbf{J}_{(m-1)(l+1)} \\
&+ \sum_{o=m}^{L-1} \mathbf{J}_{L(o+1)} \left( \phi''(\mathbf{h}_o) \circ \left[ \mathbf{W}_o \mathbf{J}_{o(m+1)} \circ \phi'(\mathbf{h}_{m+1}) \circ \mathbf{B} \mathbf{x}_m \right] \right) \frac{\partial \phi'(\mathbf{h}_o)}{\partial \mathbf{W}_m} \mathbf{W}_o \mathbf{J}_{(o-1)(l+1)}
\end{aligned} \tag{40}$$

In total, the second derivative projected into the $(\mathbf{A}, \mathbf{B})$ direction for $m > l$ is given by:

$$\begin{aligned}
\frac{\partial^2 \mathbf{x}_L}{\partial \mathbf{W}_l \partial \mathbf{W}_m} \cdot (\mathbf{A} \otimes \mathbf{B}) = \Big[ &\mathbf{J}_{L(m+1)} \phi'(\mathbf{h}_m) \mathbf{B} \mathbf{J}_{(m-1)(l+1)} + \\
&\sum_{o=m}^{L-1} \mathbf{J}_{L(o+1)} \left( \phi''(\mathbf{h}_o) \circ \left[ \mathbf{W}_o \mathbf{J}_{o(m+1)} \circ \phi'(\mathbf{h}_{m+1}) \circ \mathbf{B} \mathbf{x}_m \right] \right) \frac{\partial \phi'(\mathbf{h}_o)}{\partial \mathbf{W}_m} \mathbf{W}_o \mathbf{J}_{(o-1)(l+1)} \Big] \\
&\circ \phi'(\mathbf{h}_l) \mathbf{A} \mathbf{x}_l
\end{aligned} \tag{41}$$

Now consider the case $m = l$. Here there is no direct differentiation with respect to $\mathbf{W}_m$, but there is a derivative with respect to $\phi'(\mathbf{h}_m)$. The derivative is written as:

$$\frac{\partial^2 \mathbf{x}_L}{\partial \mathbf{W}_m \partial \mathbf{W}_m} \cdot (\mathbf{A} \otimes \mathbf{B}) = \mathbf{J}_{L(m+1)} \circ [\phi''(\mathbf{h}_m) \circ \mathbf{B}\mathbf{x}_l]\mathbf{A}\mathbf{x}_m +$$

$$\left[ \sum_{o=m}^{L-1} \mathbf{J}_{L(o+1)} \left( \phi''(\mathbf{h}_o) \circ \left[ \mathbf{W}_o \mathbf{J}_{o(m+1)} \circ \phi'(\mathbf{h}_{m+1}) \circ \mathbf{B}\mathbf{x}_m \right] \right) \frac{\partial \phi'(\mathbf{h}_o)}{\partial \mathbf{W}_m} \mathbf{W}_o \mathbf{J}_{(o-1)(m+1)} \right]$$

$$\circ \phi'(\mathbf{h}_m)\mathbf{A}\mathbf{x}_m \tag{42}$$

There are two key points: first, all but one of the terms in the off-diagonal second derivative depend on only first derivatives of the activation; for a deep network, the majority of the terms depend on $\phi''$. Secondly, on the diagonal, all terms depend on $\phi''$. Therefore if $\phi''(x) = 0$, the diagonal of the model second derivative is 0 as well.

## B  DYNAMICS OF PENALTY SAM

### B.1  PENALTY SAM VS. IMPLICIT REGULARIZATION OF SGD

The analysis of Smith et al. (2021) suggested that SGD with learning rate $\eta$ is similar to gradient flow (GF) with PUSAM with $\rho = \eta/4$. In this section we use a linear model to highlight some key differences between PUSAM and the discrete effects from finite stepsize.

Consider a quadratic loss $\mathcal{L}(\boldsymbol{\theta}) = \frac{1}{2}\boldsymbol{\theta}^{\mathrm{T}}\mathbf{H}\boldsymbol{\theta}$ for some parameters $\boldsymbol{\theta}$ and PSD Hessian $\mathbf{H}$. It is illustrative to consider gradient descent (GD) with learning rate $\eta$ and (unnormalized) penalty SAM with radius $\rho$.

The gradient descent update rule is

$$\boldsymbol{\theta}_{t+1} - \boldsymbol{\theta}_t = -\eta(\mathbf{H} + \rho\mathbf{H}^2)\boldsymbol{\theta}_t \tag{43}$$

The "effective Hessian" is given by $\mathbf{H} + \rho\mathbf{H}^2$ (see [cite ICML 2023 paper] for more analysis). Solving the linear equation gives us

$$\boldsymbol{\theta}_t = \left(1 - \eta(\mathbf{H} + \rho\mathbf{H}^2)\right)^t \boldsymbol{\theta}_0 \tag{44}$$

This dynamics is well described by the eigenvalues of the effective Hessian - $\lambda + \rho\lambda^2$, where $\lambda$ are the eigenvalues of $\mathbf{H}$. The effect of the regularizer is therefore to introduce eigenvalue-dependent modifications into the Hessian.

There is a special setting of $\rho$ which can be derived from the calculations in Smith et al. (2021). Consider $\rho = \eta/2$, and consider the dynamics after $2t$ steps. We have:

$$\boldsymbol{\theta}_{2t} = \left(1 - \eta(\mathbf{H} + \frac{1}{2}\eta\mathbf{H}^2)\right)^{2t} \boldsymbol{\theta}_0 \tag{45}$$

which can be re-written as

$$\boldsymbol{\theta}_{2t} = \left(1 - 2\eta\mathbf{H} + \eta^3\mathbf{H}^3 + \frac{1}{4}\eta^4\mathbf{H}^4\right)^t \boldsymbol{\theta}_0 \tag{46}$$

To leading order in $\eta\mathbf{H}$, this is the same as the dynamics for learning rate $2\eta$, $\rho = 0$ after $t$ steps:

$$\boldsymbol{\theta}_t = (1 - 2\eta\mathbf{H})^t \boldsymbol{\theta}_0 \tag{47}$$

We note that these two are similar only if $\eta\mathbf{H} \ll 1$. Under this condition, $\eta\rho\mathbf{H}^2 = \frac{1}{2}\eta^2\mathbf{H}^2 \ll \eta\mathbf{H}$, and the gradient penalty only has a small effect on the overall dynamics. In many practical learning scenarios, including those involving SAM, $\eta\lambda$ can become $O(1)$ for many eigenvalues during training [cite our ICML]. In these scenarios there will be qualitative differences between using penalty SAM and training with a different learning rate.

In addition, when $\rho$ is set arbitrarily, the dynamics of $\eta$ and $2\eta$ will no longer match to second order in $\eta\mathbf{H}$. This provides further theoretical evidence that combining SGD with penalty SAM is qualitatively and quantitatively different from training with a larger learning rate.

