# OpenReview forum: "How Hessian structure explains mysteries in sharpness regularization"
_ICLR.cc/2024/Conference — Submitted to ICLR 2024_

### Official Review · Reviewer_cVpc · 2023-10-31

**Soundness:** 2 fair
**Presentation:** 1 poor
**Contribution:** 2 fair
**Rating:** 5
**Confidence:** 4

**Summary:**

It is common for papers studying the Hessian of neural networks to approximate the Hessian by the so-called "Gauss-Newton matrix", which is the exact Hessian of a locally linearized model.   This paper aims to study the issues involved with making this approximation.  The paper is about aspects of deep learning that can only be explained by looking at the non-Gauss-Newton portion of the Hessian, which is termed the Nonlinear Modeling Error (NME) matrix.

The paper has two empirical findings:

 1. The authors compare SAM and Gradient Norm Penalty on both ReLU nets and GeLU nets, and find that at large $\rho$, SAM does well on both ReLU and GeLU, but GP does well only on GeLU, and performs poorly with ReLU (Figure 2).  The authors furthermore study an activation function, $\beta$-GeLU, which interpolates between ReLU and GeLU, and they show that the performance of GP continually deteriorates as the activation is made more like ReLU (Figure 3).  The authors try to explain the poor performance of GP for ReLU nets by noting that the GP gradient involves a Hessian-vector product, and the Hessian for ReLU nets is an unreliable local model for the loss landscape.  (The paper calls it a "high variance estimator" because the NME diagonals are zero at most points but nonzero at the measure-zero set of boundaries of the piecewise linear regions.)

  2.  The authors compare training with weight noise (i.e. the gradient is always computed at a gaussian-perturbed point), and the related Hessian trace penalty, to a Gauss-Newton trace penalty, and find that the latter works better, which the authors treat as evidence that it is beneficial to not penalize the NME.

---

### Post-rebuttal update:

I thank the authors for their rebuttal.

I think the experiment that is described in the general reviewer response is a good experiment.  It shows that the problem with penalty SAM on ReLU nets can be localized to the gradient of the penalty.   I would note that it is still not clear what is the mechanism by which this bad penalty gradient leads to low test accuracy (in particular, it's not clear to me that the issue can be straightforwardly interpreted as a failure to accuracy optimize the regularization term), but I suppose that this is not a claim of the paper.

Regarding the weight noise -- I understand that the paper is not interested in comparing the performance of different estimators for Hessian trace and Gauss-Newton trace.  Instead, the claim in this section (if I understand it correctly) is that an _idealized_ Hessian trace penalty (computed exactly or using infinite monte carlo samples) underperforms an _idealized_ Gauss-Newton trace penalty.  My criticism is that _this claim_ (about the idealized penalties) has not been proven convincingly.  For this reason, I think it is crucial to conduct an experiment which uses (much) more than 1 sample when computing these penalties.  This experiment would be for scientific purposes, not to be a practical algorithm.

I appreciate the clarifications about the term "second-order optimizers," as well as the clarification regarding NME vs. derivatives of NME.  The latter point should go in the paper.

Overall, due to the experiment that is described in the general reviewer response, I am feeling more confident about the paper's claims in section 4.  However, I still believe that the paper needs a lot of work (more than is appropriate to leave for the camera-ready phase) before it would meet the typical standard.  For this reason, I am raising my score to '5'.

**Strengths:**

- It's important for the field to understand what information is lost when approximating the Hessian by the Gauss-Newton matrix
- The experiments showing that gradient norm penalty ("penalty SAM") fails for ReLU networks at large rho is novel and interesting.  Understanding the cause of this experimental result is surely an important problem.
- Similarly, it's interesting that penalizing the Hessian underperformed penalizing the Gauss-Newton matrix.  Understanding the cause of this experimental result, too, is an important problem.

**Weaknesses:**

**Concerns with abstract and introduction**

The submission uses the phrase "second order methods" in a sense that is very different from the standard meaning of this phrase in the field.  I am concerned that unless this aspect of the submission is changed, most readers will take away false conclusions from this paper.   The phrase "second order methods" is usually used to refer to optimization algorithms that utilize Hessian information to converge faster.   Thus, most readers would interpret the sentence "second order methods may be more sensitive
to the choice of activation functions than first order methods" as a claim about these kinds of optimizers.  However, if I am understanding the submission correctly, the authors are **not attempting to make any claim** about these kinds of optimizers, and are instead using the phrase "second order methods" to refer to various regularizers such as a gradient norm penalty, because computing the gradient of this penalty involves computing a hessian-vector product.  If this understanding is correct, I would urge the authors to better clarify the scope of their claim.  On the other hand, if I am misunderstanding the word choice, and the authors _are_ attempting to make a claim about second-order optimization algorithms, then that claim is **completely unsupported** by the evidence in the paper.

**Concerns with section 2**

The submission says that the NME "is related to exploring the effects of switching to different linear regions."  Firstly, this statement can only possibly make sense in the context of ReLU networks or other piecewise linear activation functions like hardtanh.  Secondly, even for ReLU networks, I don't understand what this assertion means precisely.


**Concerns with section 4**

I think the claims of this section are not convincingly proven. The outlined logic in the paper would seem to go: at large rho, the gradient-norm-penalty algorithm on ReLU networks does not find "truly" flat regions, because even though it finds regions where the Hessian is small, the Hessian is a bad local model for the objective, and the objective turns upwards a bit farther away.  However, the failure of gradient norm regularization in Figure 2 (a,b) does not look like this hypothetical.  We see in this figure that the test accuracy of gradient norm penalty with large rho is smaller than that of _unregularized_ training.  This does not look like "insufficient regularization", it looks like some other kind of failure.  **Could the authors please spell out the precise claimed logical linkage between "quadratic approximation on ReLU nets gives bad local model" and the failure we see in Figure 2 (a,b)?**

There really are a lot of potential confounders here.  For large $\rho$, the dynamics of SAM and gradient norm penalty can conceivably be very different, especially at large learning rates.  For example, perhaps the issue in Figure 2 (a,b) is that ReLU units die, while GeLU units do not die?  Could you experiment with leaky-ReLU (which I believe should suffer less from the dead relu problem) to rule out this possibility?  If leaky ReLU (with a decently large leak size) breaks down at large rho, that would support your hypothesis; if it doesn't, that would support my hypothesis.

Another experiment I'd like to see would be to re-run these experiments at a considerably smaller learning rate, and to see if there is still a difference between ReLU and GeLU at large rho.  If there is still a difference, that would support your hypothesis; if there is no difference at small learning rates, that would go against your hypothesis, since (as far as I can tell) your hypothesis contains no mechanism by which learning rate would have an effect.

**Concerns with section 5**

One of the paper's main claims is that penalizing Gauss-Newton trace yields better performance than penalizing Hessian trace (which is done implicitly by weight noise).  However, I do not believe this point has been convincingly proven.  For one thing, the authors use one-sample random estimators for both Hessian trace and Gauss-Newton trace.  These one-sample estimators do not seem to be directly comparable to one another -- in one, a different random Gaussian is sampled for each parameter, while in the other, a single class label is sampled.  It seems totally plausible to me that these estimators have different properties e.g. variance that are responsible for the experimental results in this section, and that an ideal penalization of Gauss-Newton trace would not outperform an ideal penalization of Hessian trace.   I would feel much better about this claim if the authors checked how their findings hold as the number of Monte Carlo samples is made larger.  Do the findings keep holding if the authors use 2, then 5 Monte Carlo samples?  Or do the findings start to change as the number of Monte Carlo samples is made larger?

Overall, I found section 5 to be very conceptually confusing.  I had a hard time keeping track of the various claims that were being made.  As discussed above, one of the claims was clearly that an idealized penalization of Gauss-Newton trace would be better than an idealized penalization of Hessian trace (which is similar to weight noise).  However, the section _also_ seemed to be comparing these three approaches to gradient norm penalty (referred to here as "penalty SAM"). It was not clear to me what was the goal of the comparisons to GNP.  How do these comparisons relate to the paper's broader point about NME?

The introduction says:  "We show that weight noise does not perform as well as the gradient penalty it is thought to approximate."  Sorry, which gradient penalty is being referred to here?  If this sentence is referring to a _loss_ gradient norm or squared _loss_ gradient norm, then it is not correct to say that weight noise is thought to approximate this penalty.

**Questions:**

see above

---

> ### Author Response · Authors · 2023-11-16
> **Response to Reviewer cVpc**
>
> We thank the reviewer for their very detailed review. The comments have greatly aided us in improving our work.
>
> *Concerns with abstract and introduction*:
>
> Indeed, as suggested by the reviewer, our paper is not focused on second order methods which use Hessian information for optimization. We agree that many readers may believe we are referring to these methods and that was not our intention. We are editing the text to better reflect which sort of method we are attempting to analyze.
>
> *Concerns with section 2*
>
> `The submission says that the NME "is related to exploring the effects of switching to different linear regions." Firstly, this statement can only possibly make sense in the context of ReLU networks or other piecewise linear activation functions like hardtanh. Secondly, even for ReLU networks, I don't understand what this assertion means precisely.`
>
> We agree that this statement only makes sense for piecewise linear networks (ReLU, leaky ReLU, hard tanh, etc.). The idea here is that the second derivative of these networks is given by a delta function - where the location of non-zero part of the delta function is exactly the boundaries of the piecewise linear regions. This is what the NME captures in these types of models. We will improve discussion of this point in the text.
>
> *Concerns with section 4*
>
> `I think the claims of this section are not convincingly proven… at large rho, the gradient-norm-penalty algorithm on ReLU networks does not find "truly" flat regions, because even though it finds regions where the Hessian is small, the Hessian is a bad local model for the objective, and the objective turns upwards a bit farther away. However, the failure of gradient norm regularization in Figure 2 (a,b) does not look like this hypothetical. We see in this figure that the test accuracy of gradient norm penalty with large rho is smaller than that of unregularized training. This does not look like "insufficient regularization", it looks like some other kind of failure. Could the authors please spell out the precise claimed logical linkage between "quadratic approximation on ReLU nets gives bad local model" and the failure we see in Figure 2 (a,b)?`
>
> We are afraid there is a misunderstanding. For any SAM-like algorithm (SAM, gradient penalty, etc.), large values of $\rho$ eventually leads to poor training for all architectures (worse than $\rho = 0$), since practical SAM algorithms approximate the ideal SAM objective (minimax formulation at the start of 4.1). The question at hand is: why are regular SAM and penalty SAM so similar for GELU but very different for ReLU? Our hypothesis is that the structure of the NME, particularly the part relating to the second derivative of the activations, plays a role. For ReLU, the second derivative is theoretically a delta function but in practical implementations is $0$, and this implies that Hessian-vector products do a poor job of capturing local structure.
>
> `There really are a lot of potential confounders here. For large $\rho$, the dynamics of SAM and gradient norm penalty can conceivably be very different, especially at large learning rates. For example, perhaps the issue in Figure 2 (a,b) is that ReLU units die, while GeLU units do not die? Could you experiment with leaky-ReLU (which I believe should suffer less from the dead relu problem) to rule out this possibility? If leaky ReLU (with a decently large leak size) breaks down at large rho, that would support your hypothesis; if it doesn't, that would support my hypothesis.`
>
> This is an interesting hypothesis and experiment; we would like to point out that for $\beta$-gelu we also get $0$s due to machine precision, especially for larger $\beta$ (our Figure 4).
>
> We chose to do a slightly different experiment inspired by your hypothesis. We trained with ReLU, but with a custom second derivative - a Gaussian with small width centered around $0$. This gives us some approximation of the true second derivative (dirac delta function). The results are detailed in the comment to all reviewers above, but to summarize we found that adding in this approximation to the second derivative brings the accuracy of penalty SAM _above_ the baseline for a considerable range of $\rho$. This to us provides direct evidence that the second derivative (or lack thereof) is an important (but not the only!) factor in the failure of penalty SAM with ReLU. We also conducted an experiment where we _removed_ the second derivative of $\beta$-gelu, which caused penalty SAM to fall _below_ the baseline.
>
> (response cont. below)

---

> > ### Author Response · Authors · 2023-11-16
> > **Response to Reviewer cVpc (cont.)**
> >
> > `Another experiment I'd like to see would be to re-run these experiments at a considerably smaller learning rate, and to see if there is still a difference between ReLU and GeLU at large rho. If there is still a difference, that would support your hypothesis; if there is no difference at small learning rates, that would go against your hypothesis, since (as far as I can tell) your hypothesis contains no mechanism by which learning rate would have an effect.`
> >
> > One issue with this experiment is that for small learning rate, ResNet on Imagenet will not generalize well. We note that the primary effect of SAM is to improve generalization (test loss), not optimization (training loss). It is not clear what differences between ReLU and GeLU tell us at small learning rate.
> >
> > *Concerns with section 5*
> >
> > `One of the paper's main claims is that penalizing Gauss-Newton trace yields better performance than penalizing Hessian trace (which is done implicitly by weight noise). However, I do not believe this point has been convincingly proven… It seems totally plausible to me that these estimators have different properties`
> >
> > First, we overall agree with the point that we have not ruled out the efficacy of modified versions of the various penalties. There are a plethora of different ways to estimate these penalties, and indeed it is possible that different estimators (with different variances but also biases) might be useful. Our observation is that two of the most common estimation methods (Hutchinson and weight noise) don’t work well.
> >
> > One phenomenology we found (but did not have space to discuss) is that penalizing the Hessian trace often leads to large negative eigenvalues in the Hessian (via the NME). This leads to poor optimization for values of $\rho$ similar to the optimal ones for GN penalty. We hope to carry out experiments playing with the monte carlo sampling, but we won’t have time to finish those by the end of the review cycle.
> >
> > `Overall, I found section 5 to be very conceptually confusing. I had a hard time keeping track of the various claims that were being made. As discussed above, one of the claims was clearly that an idealized penalization of Gauss-Newton trace would be better than an idealized penalization of Hessian trace (which is similar to weight noise). However, the section also seemed to be comparing these three approaches to gradient norm penalty (referred to here as "penalty SAM"). It was not clear to me what was the goal of the comparisons to GNP. How do these comparisons relate to the paper's broader point about NME?`
> >
> > Your criticism is well-taken; we will edit the presentation in the text. Sections 4 and 5 are dealing with qualitatively different phenomena; Section 4 deals with the NME in the _update equation_ and Section 5 deals with the NME in the _regularizer_. We explain in more detail in the comment to all reviewers above. A quick summary is that NME in the _regularizer_ leads to a _third derivative_ in the update equation, which at least for the methods we tested does not work well. Indeed, the GN penalty involves no second derivatives in the _regularizer_ but the _update equations_ do involve second derivatives (related to the NME).
> >
> > `The introduction says: "We show that weight noise does not perform as well as the gradient penalty it is thought to approximate." Sorry, which gradient penalty is being referred to here? If this sentence is referring to a loss gradient norm or squared loss gradient norm, then it is not correct to say that weight noise is thought to approximate this penalty.`
> >
> > Here we refer to the gradient with respect to the outputs - that is, the Jacobian. We will make this more clear in the text. This is in fact an important point: the analyses in sections 4 and 5 can be cast as gradient penalties, but gradients of very different quantities (loss vs model).
> >
> > Let us know if there are any other points we can clear up.

---

> > > ### Author Response · Authors · 2023-11-22
> > > **Let us know if we can clarify any other points!**
> > >
> > > Please let us know if we can clarify anything else; if we have substantially addressed your criticisms we hope you will consider revising your review score.

---

### Official Review · Reviewer_Xtsh · 2023-11-01

**Soundness:** 4 excellent
**Presentation:** 3 good
**Contribution:** 3 good
**Rating:** 5
**Confidence:** 4

**Summary:**

While the inclusion of second-order information of the SAM method can enhance generalization, methods such as weight noise and gradient penalties often fall short in providing such benefits. This paper aims to address this issue by investigating the effectiveness of these methods in relation to the structure of the Hessian of the loss. The authors demonstrate that the Nonlinear Modelling Error matrix (NME), which has been largely overlooked in previous literature, plays a crucial role in understanding the underperformance of gradient penalties for certain architectures. Additionally, the authors present an evidence that the assumption of equivalence between weight noise and gradient penalties may not hold, as it relies on the unreasonable assumption of disregarding the NME. Drawing from their findings that regulating feature exploitation rather than feature exploration yields comparable performance to SAM, the authors emphasize the importance of appropriately controlling regularization on the two components of the Hessian for the success of various second-order methods.

**Strengths:**

This paper demonstrates the importance of the Nonlinear Modeling Error (NME) matrix, which has previously been considered insignificant due to its vanishing nature at interpolation points.

Based on the quantitative interpretation of the Hessian decomposition, they provide a new insights into the effectiveness of various regularization techniques, including SAM, weight noise, and gradient penalties. Numerical experiments illustrate their opinions.

**Weaknesses:**

The article's findings rely on empirical observations rather than theoretical proofs or extensive experiments.

Besides, the numerical comparison is only conducted for computer vision tasks. I suggest the authors to add more experiments on various network architecture to make their conclusion more convincing.

**Questions:**

When the loss function is not smooth, we can not get its second derivative. In that case, the NME matrix may not be well-defined, how can we analyze the sharpness regularization by using Hessian structures?

Can the findings on the NME matrix be used to inspire the development of the second-order methods(not the penalty regularization methods)? And how to use it?

What is Cat(z) in eq. (24)?

There are some typos in Appendix B, like ''[cite our ICML]'', ''[cite ICML 2023 paper]''.

How the conclusion are obtained ''These results are evidence that the NME term of the Hessian should not be dropped when applying
the analysis of (Bishop, 1995) to weight noise for modern networks.''? As mentioned at the beginning of section 5.2, the Hessian trace penalty consider the NME term but not perform well for both tasks. However, Gauss-Newton penalty do not consider the NME but perform better? Therefore, it seems that NME plays negative effect on this task. Could you please explain this for me?

---

> ### Author Response · Authors · 2023-11-16
> **Response to Reviewer Xtsh**
>
> We thank the reviewer for their helpful comments, and address each of them below.
>
> `The article's findings rely on empirical observations rather than theoretical proofs or extensive experiments… besides, the numerical comparison is only conducted for computer vision tasks. I suggest the authors to add more experiments on various network architecture to make their conclusion more convincing.`
>
> We point out that we did have some theoretical analysis in Section 2, which gave us the structural breakdown of the NME. This was useful to develop and explain our experiments.
>
> We have conducted additional experiments on the vision datasets; see comment to all reviewers above.
>
> `When the loss function is not smooth, we can not get its second derivative. In that case, the NME matrix may not be well-defined, how can we analyze the sharpness regularization by using Hessian structures?`
>
> The key point of our theory is that, in the case of ReLU, the second derivative is 0 pointwise almost everywhere, and is undefined at 0 (dirac delta function). In practice this second derivative is given the value 0 at the point 0 in common implementations such as in Jax. This means that for numerical methods, the NME matrix is well-defined pointwise but fails to capture the discontinuity at 0. The gradient penalty method runs but is not effective (Figure 2a/b). In contrast, for Gelu the NME correctly captures the second order information and gradient penalty method works (Figure 2 c/d).
>
> `Can the findings on the NME matrix be used to inspire the development of the second-order methods(not the penalty regularization methods)? And how to use it?`
>
> We hope to address this in future work, by co-designing activation functions, architectures, and learning algorithms.
>
> `What is Cat(z) in eq. (24)?`
>
> This is the categorical distribution with logits z. That is, we take a draw from the distribution given by softmax(z).
>
> `There are some typos in Appendix B, like ''[cite our ICML]'', ''[cite ICML 2023 paper]''.`
>
> Thanks for catching these mistakes; they have been corrected.
>
> `How the conclusion are obtained ''These results are evidence that the NME term of the Hessian should not be dropped when applying the analysis of (Bishop, 1995) to weight noise for modern networks.''? As mentioned at the beginning of section 5.2, the Hessian trace penalty consider the NME term but not perform well for both tasks. However, Gauss-Newton penalty do not consider the NME but perform better? Therefore, it seems that NME plays negative effect on this task. Could you please explain this for me?`
>
> We apologize for the confusion. The first point is that the NME is not regularized in gradient penalties - it’s involved in hessian vector products in the update rule, but the NME is not differentiated (explicitly or implicitly). See the comment to all reviewers for further discussion on this point.
>
> With regards to Bishop: the issue is that they dropped the term in _theoretical analysis_ but still regularized the term _in implementation_. In contrast, the GN penalty will drop the term *both* _analytically_ and _in implementation_. This suggests that regularizing the trace of the NME hurts training. We will edit this in the main text.
>
> Please let us know if we can clear up any other points for you.

---

> > ### Author Response · Authors · 2023-11-22
> > **Let us know if you have any followup questions**
> >
> > Please let us know if we can clarify anything else; if we have substantially addressed your criticisms we hope you will consider revising your review score.

---

### Official Review · Reviewer_Me8D · 2023-11-06

**Soundness:** 3 good
**Presentation:** 3 good
**Contribution:** 3 good
**Rating:** 6
**Confidence:** 3

**Summary:**

This paper delves into the nuances of the effectiveness of various sharpness regularization methods, such as SAM, weight noise, and gradient penalties, which are seemingly similar methods. To elucidate this, they highlight an often-overlooked term in the Hessian, named the nonlinear modeling error matrix (NME). This term, involving the second derivative of model features and activation functions (unlike the Gauss-Newton term), is proposed as a key factor in understanding these performance differences. Experiments reveal that incorporating the NME into the loss is detrimental, the choice of activation functions is crucial for gradient penalty methods, and the original SAM appears robust to such choices due to its implicit incorporation of the second-order information.

**Strengths:**

- This paper provides a plausible explanation for the varying performance of sharpness regularization methods with similar motivations, which will likely be of significant interest to the community.
- The experiments are well-designed to substantiate the paper's claims about the importance of the NME term in explaining performance discrepancies among different sharpness regularization methods.

**Weaknesses:**

- The experimental settings appear to lack sufficient robustness for the results, necessitating further justification to ensure the reliability of the findings. Notably, the experiments rely on only two seeds, and the approximation of expectations in Gauss-Newton and Hessian-trace penalty methods relies on a single sample.
- The argument regarding the relationship between feature learning and the NME term is unclear and lacks concrete support. Including intuitive examples to illustrate this connection would be helpful.

**Questions:**

- The activation Hessians do not reach an exact zero when $\beta$-GELU is used. What criteria are applied to consider the activation function Hessian as zero in Figure 4?
- Given the approximations outlined in Section 5.1, the hyperparameter $\rho$ can be aligned with specific weight noise conditions for different methods in Section 5.2. Wouldn’t it then be necessary to compare the results under these matched hyperparameters to draw more precise conclusions regarding the discrepancy between methods?
- In the case of the Hessian-trace penalty, not only $\rho$ but also the noise standard deviation in the trace estimator can be varied to control the regularization strength. Is the performance similar when both hyperparameters are controlled to achieve comparable regularization strength? If not, can the results be explained through the lens of the NME?

---

> ### Author Response · Authors · 2023-11-16
> **Response to reviewer Me8D**
>
> We thank the reviewer for their response, and attempt to address some of their concerns below.
>
> `The experimental settings appear to lack sufficient robustness for the results, necessitating further justification to ensure the reliability of the findings. Notably, the experiments rely on only two seeds, and the approximation of expectations in Gauss-Newton and Hessian-trace penalty methods relies on a single sample.`
>
> We are confident that the results will hold up over more seeds given the magnitude of the effects compared to the error; we plan to run these experiments before the camera ready. With regards to the stochastic approximation of the GN and Hessian trace: we agree that this will lead to estimation error. We did not exhaustively explore estimation schemes because that is beyond the scope of this work. We used a single sample because that is common in the use of weight noise and often important in terms of efficiency. Please also see the "comment to all reviewers" for additional experiments.
>
> `The argument regarding the relationship between feature learning and the NME term is unclear and lacks concrete support. Including intuitive examples to illustrate this connection would be helpful.`
>
> We appreciate this concern. The intuition is that the change in the Jacobian matrix $J$ is controlled by the second derivative $\nabla_{\theta}^{2} z$; this term is present in the NME (but not in GN). On the other hand, The Jacobian $J$ is related to the features in the sense of the neural tangent kernel (NTK) which can be computed as $J J^T$. We will add this point to the main text.
>
> `The activation Hessians do not reach an exact zero when $beta$-GELU is used. What criteria are applied to consider the activation function Hessian as zero in Figure 4?`
>
> Thanks for catching this subtle point; because we use finite precision, we reach floating point 0 even though the derivative is not actually 0. In particular $\beta$-gelu goes as $\exp(-(\beta x)^2)$ for large $x$, so numerical $0$ is reached quite easily.
>
> `Given the approximations outlined in Section 5.1, the hyperparameter $\rho$ can be aligned with specific weight noise conditions for different methods in Section 5.2. Wouldn’t it then be necessary to compare the results under these matched hyperparameters to draw more precise conclusions regarding the discrepancy between methods?`
>
> We cross-validated over parameters because all the methods differ in nontrivial ways - they estimate different quantities, with different errors and biases. The cross-validation ensures we don’t miss a well-performing setting for any method.
>
> `In the case of the Hessian-trace penalty, not only $\rho$ but also the noise standard deviation in the trace estimator can be varied to control the regularization strength. Is the performance similar when both hyperparameters are controlled to achieve comparable regularization strength? If not, can the results be explained through the lens of the NME?`
>
> The Hessian trace is worse than the weight noise at all matching hyperparameter values. We omitted detailed discussion due to space constraints, but the Hessian trace penalty can rapidly drive large negative eigenvalues in the NME.
>
> Let us know if there are any other questions we can answer.

---

> ### Author Response · Authors · 2023-11-22
> **Any followup questions?**
>
> Please let us know if we can clarify anything else! If we have substantially addressed your criticisms we hope you will consider revising your review score.

---

> > ### Comment · Reviewer_Me8D · 2023-11-23
> >
> > I appreciate the authors' responses. I will keep my rating.

---

### Official Review · Reviewer_jCBc · 2023-11-07

**Soundness:** 2 fair
**Presentation:** 1 poor
**Contribution:** 1 poor
**Rating:** 3
**Confidence:** 4

**Summary:**

This work analyzes why imposing the gradient norm penalty does not work as well as using SAM, despite the two matching in objectives when the loss is approximated up to the first order. The paper shows how this issue does not seem to occur with GeLU as opposed to ReLU, and then attributes it to ReLU's almost zero everywhere curvature which, as a result, does not get properly manifested in the indefinite part of the Hessian. This is then somewhat corroborated by interesting evidence on ImageNet and CIFAR10. Lastly, a similar perspective is attempted in order to understand and compare the weight noise regularization as opposed to other methods.

**Strengths:**

- Interesting perspective that would emphasize the need to take additional care when accessing second-order information for networks with ReLU activation
- Help reconcile the seemingly worse behaviour of various gradient penalty algorithms. More generally, by studying gradient penalty methods, further insights could be achieved into the inner workings of sharpness-regularizing algorithms and the loss landscape at large.

**Weaknesses:**

- **Feature exploitation and exploration analogy is interesting but also crude:** Sure, the second term in the Hessian tells more directly about feature exploration than the GN part. But the GN term during training is not fixed, but evolves. And its evolution or change is naturally dependent on the second term in the Hessian. Generally speaking, the linearization argument isn't fully correct either, as that inherently considers a fixed Hessian (like linearize around initialization or about some other point). Consequently, the whole analogy is rather crude and rife with ambiguities.

&nbsp;

- **Explanation of gradient penalty via the activation function:** I do agree that this could be a valuable perspective in general, and indeed the various experimental results do offer a partial hint towards what may be the core issues that impede the efficacy of using gradient norm penalty for networks with ReLU. Yet, I don't think the argument is convincing enough to fully attribute this to the nature of the activation function. (a) The off-diagonal terms in the NME will still have a highly non-trivial contribution, and these are present regardless the almost everywhere zero entries on its diagonal. There is not much of a quantification as to the precise significance of the diagonal entries of the NME, and in what manner it influences the overall Hessian. (b) The results in Figure 3 seem somewhat misleading for they do not quite match the value corresponding to ReLU ($\beta \rightarrow \infty$) shown in Figure 2. For instance, with $\rho=0.1$, the accuracy in ImageNet and CIFAR10 for PSAM with ReLU seems to be about 69% and 94% from naked eye. But that in Figure 3 seems to be much smaller. This weakens the presented argument.

&nbsp;

- **Weight noise does not work:** This section as said elsewhere is poorly written, and perhaps due to that or in general, comes across as a bit disconnected, and is also not properly fleshed out. I do get more or less the whole pretext around this, but the execution is lacking. The "roughly matches" the results of the Hessian trace penalty is a bit too rough. Besides, how are the results in the case of ReLU? Right now, with the fairly decent results, I would also not claim that weight noise does not work.

&nbsp;


- **Wrong citations and a general sloppiness in discussing related work:**

   - Moosavi-Dezfooli et al. 2019 citation about regularizing the Hessian to improve generalization is an incorrect in this context. They are talking about Hessian with respect to the inputs, not parameters as is the focus of this paper. It kind of betrays the rigour (or lack of it) when such a thing happens on the second citation of the paper.
   - The more accurate citation for USAM is Andriushchenko & Flammarion (2022),  not what it currently comes across to be as Agarwala & Dauphin 2023.
   - "The second term to our knowledge does not have a name so we call it the Nonlinear Modeling Error matrix": It has been called functional Hessian in the works of Singh et al., 2021, 2023 analyzing Hessian rank for instance, --- where this matrix has also been analyzed in theoretical detail.
   - Many second-order optimizers approximate Hessian only with Gauss-Newton: It would make the discussion balanced by adding that the reason for doing so is to also  ensure guaranteed decrease in loss at each step (through  a PSD precondition).

&nbsp;


- **Imprecise writing:**
    - Section 5 onwards the quality of writing drops, and there is often a lack of coherence across the neighboring paragraphs. It seems very hastily written.
    -  I don't think it's a great idea to introduce sections by saying the key idea of this paper is to explain the phenomenon discussed in some other section 3. "The key hypothesis of this paper is that the structure of the Hessian can be used to explain the empirical phenomena of Sections 4 and 5."
    - "rescue the performance of gradient penalty": really 'rescue'? :)
   - 'mysteries': i don't think the touched upon issues are so grave or so mind-bending to be qualified as mysteries. all it serves right now is to purposefully and unnecessarily lend it a 'mysterious' air. we all could also use some earnestness when naming our papers.

----

I would have liked to give a better score, but the crucial issues above, plus a somewhat insufficient contribution (section 5 only dilutes, not supplements the analysis before), makes me inclined towards rejection.

**Questions:**

Please look at the Weaknesses section. Besides, I have some other minor questions:

- Can you add the training accuracy for the plots in Figure 2?
- How is the gradient norm penalty exactly implemented? I guess using a mini-batch? Of the same size as that for SAM?
- Figure 4: Is this computed across all neurons? Over the entire dataset?
- The Hessian trace is approximated over how many samples?

---

> ### Author Response · Authors · 2023-11-16
> **Response to reviewer jCBc**
>
> We thank the reviewer for their very thorough response; their feedback will help us improve this work tremendously.
>
> With regards to the explore/exploit analogy: we agree that the analogy is not precise. We included it to give some insight into the differences between the GN and NME, at least in the case of ReLU and similar activations. The analogy refers to what information the GN and NME immediately surface; as pointed out by the reviewer, the time evolution of the GN depends on the second derivative of the model with respect to parameters, and therefore is related to the NME.
>
> `The off-diagonal terms in the NME will still have a highly non-trivial contribution, and these are present regardless the almost everywhere zero entries on its diagonal. There is not much of a quantification as to the precise significance of the diagonal entries of the NME.`
>
> We agree that the off-diagonal term is still present in ReLU, and may in fact contribute to learning. However we can test the effects of the second order terms directly. We defined an “augmented ReLU” function whose second derivative was a Gaussian (approximation of the true second derivative, a 0-width dirac delta). The details of the experiment are in the comment to all reviewers above; the upshot is that adding this second derivative term causes penalty SAM to achieve higher accuracy than baseline for a non-trivial range of $\rho$. We also conducted an experiment where we set the second derivative of gelu to be $0$ (diminished gelu). Removing this second derivative led to penalty SAM performing _below_ the unregularized baseline.
>
> We thank the reviewer for noticing the apparent discrepancy in Figure 3. This is due to the fact that at finite $\beta$ the Beta-Gelu has second derivative $\beta$ for input $0$, while the ReLU implementation in Jax has second derivative $0$ for input $0$. (This is a common choice of second derivative for ReLU.) This leads to some differences in the training dynamics between large $\beta$, large $\rho$, and ReLU + large $\rho$. We will clarify this in the paper.
>
>
> We will add/fix the suggested references, and improved our overall discussion of the references. We have also improved our overall discussion of second order methods, and have noted that PSD-ness is useful to have decreasing loss for preconditioning methods (at small learning rate).
>
> We appreciate the criticism that Section 5 is confusing. We posit that the lack of success of weight noise is well known in the community, and indeed in our own analysis weight noise does not lead to significant improvement over SGD (0.3% accuracy increase over SGD on Imagenet, with error of 0.1% over 2 samples; 0.1% increase with error 0.1% on CIFAR10). We acknowledge the issues with the presentation of this section. We have provided additional insight into section 5 in our comment to all reviewers, and we are in the process of improving the discussion in the main text.
>
> Minor questions:
>
> `Can you add the training accuracy for the plots in Figure 2?` We will add this.
>
> `How is the gradient norm penalty exactly implemented? I guess using a mini-batch? Of the same size as that for SAM?` Yes, exactly - gradient is computed on one minibatch, then autodiff is used to differentiate the gradient norm on the same minibatch.
>
> `Figure 4: Is this computed across all neurons? Over the entire dataset?` This is computed across all activations over a minibatch of 1024 samples.
>
> `The Hessian trace is approximated over how many samples?` Hessian trace is computed over the minibatch, which is 1024 samples.
>
> Please let us know if we can answer any more questions; happy to clarify any points we made above.

---

> > ### Author Response · Authors · 2023-11-22
> > **Any further questions?**
> >
> > Please let us know if we can clarify anything else; if we have substantially addressed your criticisms we hope you will consider revising your review score.

---

### Author Response · Authors · 2023-11-16
**Comment to all reviewers**

We thank all of the reviewers for their comments; we will address some points relevant to multiple reviewers here.

One critique of section 4 was that there wasn’t enough evidence to support the claim that the second derivatives in particular were driving the phenomenology. We conducted two additional experiments (ImageNet) to address this point.

In the first experiment, we “added” second derivative to the ReLU network in the following way. The second derivative of ReLU is, in principle, the dirac delta function, but in most implementations it is 0 everywhere. We replaced this derivative with a Gaussian of variance $1/\beta$. In the limit of large $\beta$, this Gaussian better and better approximates the delta function. However for any fixed $\beta$, the ReLU now has a second derivative which contributes to the NME term. The idea here is that there is some tradeoff between the goodness of the approximation to the delta function, and the amount of second derivative added.

We found that for $\beta = 10$, this “Augmented ReLU” showed improvement over the baseline for penalty SAM with $\rho$ as large as $0.05$. This is in contrast to normal ReLU, where for $\rho\geq 0.01$ the baseline performs better than penalty SAM. This suggests that adding a second derivative to ReLU can improve the performance of penalty SAM. The results are summarized below.

$\rho$ |  ReLU  | Augmented ReLU

0.00         | 76.80    | 76.80

0.01    | 77.15    | 77.69

0.05    | 74.01    | 77.42

0.10      | 69.47    | 67.48

In the second experiment, we took GELU and set its second derivative to $0$ (diminished GELU). This _removes_ the second derivative part of the NME from the penalty SAM calculations. Removing the second derivative causes penalty SAM to achieve an accuracy _below_ the baseline for $\rho > 0.01$. This suggests that removing second derivative information from the NME hurts penalty SAM.

$\rho$ |  GELU  | Diminished GELU

0.00         | 77.22    | 77.02

0.01    | 77.56    | 77.43

0.05    | 78.66    | 73.60

0.10      | 79.14    | 67.37

Taken together, our experiments suggest that second derivative information in the NME has a significant beneficial impact on penalty SAM training. We note that in both the experiments, the custom second derivative is only called during Hessian vector products. The first derivative is never affected.

Another common question was about the relationship between sections 4 and 5, and the seeming lack of consistency in the results. We would like to clarify some points here.

There are two forms of NME that we study in our work:

*NME in the update equations.* (Section 4.) This is where the NME comes up in gradient penalty regularizers - the update step involves a Hessian-vector product, which involves the NME. In this case, we found that the second derivative information in the NME helps learning dynamics.

*NME in the regularizer.* (Section 5.) Here the NME is in the regularizer itself. This means that the update step involves the _derivative of the NME_ (which involves _third derivatives_). We show that the impact of the NME cannot be ignored as suggested by (Bishop, 1995) and in fact penalizing its trace leads to poor outcomes.

We are working to include this discussion in the text.

---

### Meta-Review · Area_Chair_XFCj · 2023-12-08

**Metareview:**

The work sheds light on how the two components of the loss Hessian--the Gauss Newton (GN) term and the `nonlinear modeling error' (NME) term--impacts sharpness regularization. The work tries to demonstrate the impacts of ignoring the NME matrix, benefits of penalizing the GN matrix, among other observations.

The reviewers appreciate the direction pursued by the work and the responses (and additional experiments) provided. Based on the feedback from the reviewers -- since the work is largely empirical, the case may be more convincing with additional experiments, some of which have been proposed+discussed by the reviewers. The work is making rather intriguing claims about the behavior of the NME and the work will be stronger if there is stronger empirical evidence to support the claim, e..g., better estimate of the NME, different types of datasets, etc.

I do think, as do some of the reviewers, that the authors are on to something major. The final version of these results will likely impact both practice and theory of deep learning optimization. Given its potential impact, we do hope the authors do additional experiments to support more convincing evidence of the claims.

**Justification For Why Not Higher Score:**

The work pursues an intriguing direction, and will benefit from more convincing empirical evidence supporting some of the key claims.

**Justification For Why Not Lower Score:**

N/A

---

### Decision · Program_Chairs · 2024-01-16

Reject